# A Novel Urinary miRNA Biomarker for Early Detection of Colorectal Cancer

**DOI:** 10.3390/cancers14020461

**Published:** 2022-01-17

**Authors:** Hiroyasu Iwasaki, Takaya Shimura, Mika Kitagawa, Tamaki Yamada, Ruriko Nishigaki, Shigeki Fukusada, Yusuke Okuda, Takahito Katano, Shin-ichi Horike, Hiromi Kataoka

**Affiliations:** 1Department of Gastroenterology and Metabolism, Nagoya City University Graduate School of Medical Sciences, 1 Kawasumi, Mizuho-cho, Mizuho-ku, Nagoya 467-8601, Japan; hiwasaki@med.nagoya-cu.ac.jp (H.I.); mk313659@med.nagoya-cu.ac.jp (M.K.); nishiga2@med.nagoya-cu.ac.jp (R.N.); a100405@med.nagoya-cu.ac.jp (S.F.); okuda10@med.nagoya-cu.ac.jp (Y.O.); takatano@med.nagoya-cu.ac.jp (T.K.); hkataoka@med.nagoya-cu.ac.jp (H.K.); 2Okazaki Public Health Center, 1-3 Harusaki, Harisaki-cho, Okazaki 444-0827, Japan; t-yamada@okazaki-med.or.jp; 3Advanced Science Research Center, Kanazawa University, 13-1 Takaramachi, Kanazawa 920-8640, Japan; sihorike@staff.kanazawa-u.ac.jp

**Keywords:** colorectal cancer, biomarker, urinary miRNA, miR-129-1-3p, miR-566

## Abstract

**Simple Summary:**

Early diagnosis is critically important to achieve life-saving therapy for colorectal cancer (CRC). Since colonoscopy is not suitable as a screening method for CRC due to its invasiveness and high-cost, reliable and non-invasive diagnostic biomarkers are hopeful for CRC. In this case-control study, we established completely non-invasive, novel urinary microRNA (miRNA) biomarker panel combining miR-129-1-3p and miR-566 for the diagnosis of CRC. In the independent age- and sex-matched three cohorts comprising 415 participants, urinary levels of these miRNAs were consistently elevated in the CRC group compared to the healthy controls. Notably, the panel of combining miR-129-1-3p and miR-566 revealed an AUC of 0.845 for stage 0/I CRC that can be treated with endoscopic resection.

**Abstract:**

Since noninvasive biomarkers as an alternative to invasive colonoscopy to detect colorectal cancer (CRC) are desired, we conducted this study to determine the urinary biomarker consisting of microRNAs (miRNAs). In total, 415 age- and sex-matched participants, including 206 patients with CRC and 209 healthy controls (HCs), were randomly divided into three groups: (1) the discovery cohort (CRC, *n* = 3; HC, *n* = 6); (2) the training cohort (140 pairs); and (3) the validation cohort (63 pairs). Among 11 urinary miRNAs with aberrant expressions between the two groups, miR-129-1-3p and miR-566 were significantly independent biomarkers that detect CRC. The panel consisting of two miRNAs could distinguish patients with CRC from HC participants with an area under the curve (AUC) = 0.811 in the training cohort. This panel showed good efficacy with an AUC = 0.868 in the validation cohort. This urinary biomarker combining miR-129-1-3p and miR-566 could detect even stage 0/I CRC effectively with an AUC = 0.845. Moreover, the expression levels of both miR-129-1-3p and miR-566 were significantly higher in primary tumor tissues than in adjacent normal tissue. Our established novel biomarker consisting of urinary miR-129-1-3p and miR-566 enables noninvasive and early detection of CRC.

## 1. Introduction

Colorectal cancer (CRC) is a frequent cause of cancer deaths worldwide [1]. Because early-stage CRC is curable by minimally invasive therapy in many cases, early detection through mass screening is important for reducing mortality. The gold standard for CRC diagnosis is pathological diagnosis using biopsy samples obtained through a colonoscopy (CS). Although CS screening shows high CRC detection rates and adenomas [2], it has not been widely applied for screening tests due to its invasiveness and high cost. The fecal immunochemical test (FIT) has been established as a widely recommended screening tool to detect CRC [3,4]. However, it has been reported that 20–40% of CRC, especially stage 0/I CRC, is not detectable by FIT [5,6,7]. In recent years, the multitarget stool DNA screening test that detects CRC-related genetic mutations has been developed to detect CRC [8,9]; however, there is insufficient evidence to warrant its replacement of FIT. In addition, it is challenging to handle stool samples because of bacterial abundance, odor, and contamination of food residues. Although serum tumor markers, such as carcinoembryonic antigen (CEA) and carbohydrate antigen 19-9 (CA19-9), are often used as noninvasive CRC markers during medical checkups, they are inappropriate as screening tools due to their low sensitivity, especially for early disease [10,11,12]. It is thus important to establish a noninvasive biomarker for the early diagnosis of CRC.

MicroRNAs (miRNAs) are short non-coding RNAs that regulate the expression of target genes through messenger RNA degradation. Their aberrant expression seems to be involved in carcinogenesis [13,14]. Because miRNAs form complexes with Argonaute proteins, some lipids, and microvesicles when transported [15], they are protected from degradation and considered relatively stable under various storage conditions [16,17,18,19]. Therefore, they should act as biomarkers. Although many researchers have reported diagnostic biomarkers for CRC using serum or plasma miRNAs [20,21], there are no known biomarkers consisting of urinary miRNAs [22]. Urine is an ideal sample for medical checkups because of its noninvasiveness, easy handling, and low cost. We have made a longstanding effort to discover urinary biomarkers and established urinary protein biomarkers for diagnosing gastric cancer (GC) and CRC [23,24,25,26]. Moreover, we have also identified the urinary miRNA biomarker to detect GC [27] and esophageal cancer (EC) [28]. Based on this background, we conducted this study to establish reliable and noninvasive urinary miRNA biomarkers for CRC. 

## 2. Materials and Methods

### 2.1. Patients and Study Design

We studied 522 urine samples from 223 patients with CRC and 299 healthy controls (HCs). All samples were collected from September 2012 to August 2018 at three Japanese institutions. We included males and females aged 20–90 years. Patients with CRC (CRC group) had an existing cancer diagnosis, established by histological and endoscopic findings, and no prior treatment on entry. HCs were recruited from healthy individuals without any symptoms, and had no neoplasms as confirmed by a medical checkup. Individuals with previous cancer or other malignancies within the past 5 years were excluded from the study. There were no criteria for timing of urine collection and for preparation before urine collection. To ensure the accuracy and comprehensiveness of reporting in this case-control biomarker study, we complied with both the REMARK guidelines [29] and the STROBE statement [30]. This study was registered with the University Hospital Medical Information Network Clinical Trials Registry (UMIN000021350).

### 2.2. Samples and Definition

Urine samples were collected from each patient with CRC before any treatment and immediately stored at −80 °C until analyzed, as reported previously [23,24,25,27]. All patients with CRC were classified based on Tumor Node Metastasis staging and the Union for International Cancer Control guidelines, version 7 [31].

### 2.3. miRNA Extraction

The procedure was described in a previous report [27]. Briefly, 200 µL (600 µL for microarray use) of urine or serum was used for extracting miRNA by miRNeasy Serum/Plasma Kit (Qiagen, Valencia, CA, USA) according to the manufacturer’s instructions. Extraction of miRNAs from formalin-fixed paraffin-embedded (FFPE) tissues was conducted using the miRNeasy FFPE Kit (Qiagen). 

### 2.4. miRNA Microarray Assay

The miRNA microarray assay was conducted as described in a previous report [27]. Briefly, Cyanine-3 (Cy3) labeled cRNA were synthesized using the miRNA Complete Labeling and Hyb Kit (Agilent, Santa Clara, CA, USA) according to the manufacturer’s protocol. Cy3-labeled miRNA specimens were hybridized to the Agilent Human miRNA Microarrays (G4872A). After overnight for hybridization, microarrays were scanned by the Agilent DNA Microarray Scanner (G2539A). The obtained images were analyzed with Feature Extraction Software 11.0.1.1 (Agilent). 

### 2.5. Quantitative Reverse Transcription-Polymerase Chain Reaction (qRT-PCR)

The protocol was also described previously [27]. Briefly, complementary DNA (cDNA) was prepared from miRNA samples using TaqMan Advanced MicroRNA cDNA Synthesis Kit (Applied Biosystems, Foster, CA, USA), according to the manufacturer’s instructions. Quantitative PCRs were conducted in duplicate using the TaqMan Advanced MicroRNA Assay (Applied Biosystems) and TaqMan Fast Advanced Master Mix (Applied Biosystems) by 7500 Fast Real-Time PCR system (Applied Biosystems). We calculated cycle threshold (Ct) values to quantify miRNA expression using the 2^−∆Ct^ method. Internal controls for normalization in qPCR of urinary miRNA were determined using a global mean normalization method with the microarray results [32]. Therefore, miR-4669 and miR-6756-5p were determined to be the internal normalization controls for qPCR of urinary and serum miRNAs, as shown in the previous study [28]. As the internal normalizer for the qPCR of miRNA in FFPE tissues, we used RNU6B. The reagents used in qRT-PCR were listed in Appendix A.

### 2.6. In Silico Analyses

Kaplan–Meier curves showing the relationship between miRNA expression and survival time of the patients of rectal adenocarcinoma was downloaded from Kaplan–Meier Plotter (https://kmplot.com/analysis/index.php?p=service&cancer=pancancer_mirna (accessed on 17 November 2021)).

### 2.7. Statistical Analyses

Matching between the CRC and HC groups was conducted using a propensity score (PS) determined by a logistic regression model (age and gender). The two groups were randomly matched one-to-one using the nearest-neighbor method within a caliper width of 25% of the standard deviation of the PS logit. 

The Mann–Whitney U test, Student’s *t*-test (for serum creatinine values), and chi-squared test were used for detection of the significant differences as appropriate. We evaluated correlation using Spearman’s rank method with a coefficient (r). Receiver operating characteristic (ROC) curve analysis was used to calculate the area under the curve (AUC) for each biomarker, and the AUC value with a 95% confidence interval (CI) was shown as the representative value. Logistic regression modeling was used to estimate the odds ratio (OR) with 95% CI and construct a formula for scoring, which, in turn, was used to draw the ROC curve to compute the AUC for the combination biomarker. Instead of the actual measured values, the Z score’s adjusted values were used to calculate OR. Statistical analyses were carried out using R software (https://www.R-project.org/, (accessed on 17 November 2021)) or IBM SPSS statistics, version 25 (IBM Corp., Tokyo, Japan), respectively. All *p* values were two-sided, and those <0.05 were considered statistically significant.

## 3. Results

### 3.1. Participants

The study flowchart is shown in Figure 1. Among 522 participants comprising 223 patients with CRC and 299 HC subjects, 415 age- and sex-matched participants were enrolled in the study (206 patients from the CRC group and 209 participants from the HC group). Afterward, this cohort was randomly divided into three groups, with nine participants (three patients from CRC group and six participants from HC group) in the discovery cohort, 280 participants (140 pairs) in the training cohort, and 126 participants (63 pairs) in the validation cohort. There were no significant differences for all factors between the two groups. About two-thirds of CRC group had sigmoid or rectal cancer, and 66 patients (32.0%) with CRC had stage 0 or I CRC (Table 1).

### 3.2. Urinary miRNA Difference between HC and CRC Groups

First, to detect differences in urinary miRNAs between the HC and CRC groups comprehensively, we conducted an miRNA microarray analysis in the discovery cohort (HC = 6 vs. CRC = 3). Eleven urinary miRNAs showed significantly aberrant expressions between the HC and CRC groups (Appendix A). 

### 3.3. Development of Urinary miRNA Biomarker

Among 11 candidate miRNAs identified through microarray analysis, eight miRNAs revealed unstable urine sample expression. Consequently, we quantitated three miRNAs using qRT-PCR in the next training cohort. 

Univariate analysis showed that urinary expression levels of miR-129-1-3p, miR-566, and miR-598-5p were significantly higher in the CRC group than in the HC group (*p* < 0.001). Moreover, multivariate analysis revealed that urinary levels of miR-129-1-3p (OR: 5.59 [95% CI, 2.82–11.10]; *p* < 0.001) and miR-566 (OR: 1.64 [95% CI, 1.09–2.45]; *p* = 0.017) were also independent biomarkers for the diagnosis of CRC (Table 2). Based on these results, we established a diagnostic biomarker panel of CRC consisting of urinary miR-129-1-3p and miR-566 using a logistic regression model. This urinary miRNA biomarker panel showed satisfactory power to distinguish patients with CRC from HC participants with an AUC = 0.811 (95% CI, 0.762–0.861), which was higher than that of either miR-129-1-3p or miR-566 alone (Figure 2A). When the cut-off point was determined at the Youden index, this logistic regression model showed good efficacy, with 80.7% sensitivity, 70.7% specificity, and 75.7% accuracy for detecting CRC.

### 3.4. Validation of Urinary miRNA Biomarker

Further validation of this diagnostic biomarker panel was performed in an independent cohort (the validation cohort) to ensure extrapolation. Urinary expression levels of both miR-129-1-3p and miR-566 were significantly higher in the CRC group than in the HC group (*p* < 0.001). These results were consistent with findings in the training cohort (Table 3). The combination biomarker panel also showed a good AUC = 0.868 (95% CI, 0.806–0.931) with 88.9% sensitivity, 76.2% specificity, and 82.5% accuracy in the validation cohort (Figure 2B). 

Since background factors may affect urinary miRNA expression, we investigated the relationship between urinary levels of these miRNAs and clinical parameters (age, gender, degree of differentiation, and serum creatinine level). Although age, degree of differentiation, and serum creatinine level were not correlated with the expression of urinary miRNAs, both urinary miRNAs were significantly higher in female than in male (Appendix A). However, gender was not significant upon the multivariate analysis, and our established urinary miRNA biomarker panel showed good efficacy in both the male and female cohorts with an AUC = 0.882 (95% CI, 0.839–0.925) and 0.773 (95% CI, 0.703–0.843), respectively (Appendix A). 

Next, we investigated the diagnostic ability for early-stage CRC. In a comparison between HC participants and patients with early-stage CRC, both urinary miR-129-1-3p and miR-566 showed significantly higher levels in the stage 0/I CRC group than in the HC group (*p* < 0.001) (Figure 3A). This urinary miRNA biomarker panel also showed excellent power to distinguish patients with stage 0/I CRC from HC participants with an AUC = 0.845 (95% CI, 0.798–0.893) (Figure 3B). Conversely, expression levels of both miR-129-1-3p and miR-566 in urine did not correlate to the disease stage (Appendix A). In Kaplan–Meier curves based on the Kaplan–Meier Plotter, both miR-129 (logrank *p* = 0.37) and miR-566 (logrank *p* = 0.21) had no correlation with overall survival of the patients with rectal adenocarcinoma. Regardless of disease stage, this urinary biomarker was superior to currently used tumor markers (serum CEA and CA19-9) (Appendix A). Of note, this urinary combination biomarker panel showed 82.8% sensitivity for stage 0/I CRC, whereas both serum CEA and CA19-9 showed only 11.1% sensitivity for stage 0/I CRC. These results suggest that our established urinary biomarker panel is a useful noninvasive screening tool for the early detection of CRC.

In addition, regardless of the degree of differentiation, these urinary miRNA biomarkers showed significantly higher expression levels in the CRC group than in the HC group (Appendix A). 

### 3.5. Analysis Using Serum and Tissue Samples

Next, we analyzed serum levels of miR-129-1-3p and miR-566, but no significant correlations were found between urine and serum levels. Nevertheless, serum miR-566 showed a significantly higher expression level in the CRC group than in the HC group, and serum miR-129-1-3p showed the same tendency (Appendix A). Because it is unclear whether urinary miR-129-1-3p and miR-566 are derived from CRC tissues, we also measured the expression levels of these miRNAs in tissue samples. Interestingly, expression levels of both miR-129-1-3p and miR-566 were significantly higher in the primary tumor tissues than in the adjacent normal tissues (Appendix A). Appendix A shows the characteristics of patients with CRC for tissue miRNA analysis.

## 4. Discussion

This large sample study, including three independent cohorts, clearly showed that the urinary biomarker panel combining miR-129-1-3p and miR-566 is a novel diagnostic noninvasive biomarker to detect CRC, even at an early stage. While the sensitivity of FIT for detecting advanced adenoma (i.e., stage 0 CRC) was reportedly 11–56% [5,6,7], our urinary biomarker showed good sensitivity of 82.8% for detecting stage 0/I CRC patients. Although we cannot simply compare the two methods, our established urinary miRNA biomarker might overcome FIT in point of early detection of CRC.

Urine is an ideal sample for mass screening because of its easy handling and collection. Previous studies have reported the usefulness of urinary methylated or mutated genes for CRC detection [22]; however, additional investigation is needed because of low sensitivity and lack of information for early detection. Moreover, such genetic markers may not be able to pass through the glomerulus because of their high molecular weight [33]. Because miRNAs are small molecules consisting of 20–25 nucleotides, miRNAs have an advantage as urinary biomarker targets. Indeed, miR-129-1-3p and miR-566 expression levels were also elevated in both serum and tissue samples in this study, suggesting that the overexpression of these miRNAs in CRC tissues would be secreted into the serum and finally excreted into the urine. These results were consistent with our previously identified urinary miRNA biomarkers for GC and EC [27,28]. No direct correlations were found for the two urinary miRNAs between urine and serum levels. Similarly, the previous study reported different miRNA profiles between plasma and other fluids [34]. We also showed different signatures between serum and urinary miRNAs in the previous biomarker study of EC [28]. These results have suggested that serum miRNAs might be susceptible to other abundant factors in serum, and the present miRNA urinary biomarkers revealed ideal performance of predictability for the presence of CRC in urine through the selective filtering process.

There are several reports using serum/plasma and fecal miRNAs to detect CRC. One study showed that the expression level of serum miR-1290 is increased in patients with CRC and distinguished patients with CRC efficiently [21]. Another study indicated that the diagnostic panel consisting of fecal miR-421, miR-27a-3p, and hemoglobin showed better efficacy than hemoglobin alone [35]. However, there are no known reports of urinary miRNA biomarkers to detect patients with CRC. Because urine contains only small amounts of miRNA compared with serum/plasma and feces, it would appear to be difficult to detect the small difference in urinary miRNA expression. However, it is also a strong advantage that urine contains very few substances such as bacteria and protein that cause some expression analysis noises. In addition, urinary miRNAs seem stable under various storage conditions [17,36].

miR-129 family members are generally considered tumor suppressors with decreased expression in various cancers, which often refers to miR-129-5p [37,38,39]. In terms of miR-129-1-3p, its downregulation was associated with tumor progression via the c-Src pathway in CRC cells and tissues [40], in opposition to our results. Another study indicated that miR-129-1-3p promoted cell proliferation via programmed cell death in GC cells [41]. In our study, miR-129-1-3p was overexpressed in CRC tissues, but its inhibition did not affect cell proliferation and migration (data not shown). Although the function of miR-129-1-3p as related to CRC is controversial, overexpressed miR-129-1-3p may be a result accompanied with carcinogenesis and not a key oncogenic driver.

miR-566 stimulates epidermal growth factor receptor pathway via von Hippel–Lindau disease [42]. In addition, another study reported the oncogenic behavior of miR-566 in renal cell carcinoma (RCC) and upregulated expression in RCC tissues and cells [43]. These conclusions support our findings that miR-566 was upregulated in CRC tissues. Conversely, other studies showed that reduced expression of miR-566 was correlated to CRC development [44] and was involved in epithelial–mesenchymal transition (EMT) driven by *Alu* RNA [45]. Our study showed the urinary level of miR-566 was elevated even in the patients with early-stage CRC, which is generally unrelated to EMT, suggesting that miR-566 may be involved in the carcinogenesis of CRC in complicated ways.

This study has two limitations. First, the oncogenic function of miR-129-1-3p and miR-566 in CRC remains unknown. However, the significant results reported here were consistent among the three independent cohorts. Furthermore, our established urinary miRNA biomarker could efficiently detect stage 0/I CRC, which was a major advantage for developing a mass screening tool. Notably, this urinary miRNA biomarker could show a much higher detection rate than the serum tumor markers currently used. Although additional basic studies are also needed to clarify these miRNAs’ mechanism as related to CRC, we think that our urinary miRNA biomarker could be the next-generation CRC screening test. Second, we need to set an optimal cut-off value for the future clinical use. A prospective study is essential to validate the efficacy of this urinary biomarker and set a cut-off with well-balanced sensitivity and false positive rate. We are thus planning the prospective cohort study for the future clinical application.

## 5. Conclusions

In conclusion, a novel urinary biomarker consisting of miR-129-1-3p and miR-566 has made it possible for the early detection of CRC in a completely noninvasive manner.

## Figures and Tables

**Figure 1 cancers-14-00461-f001:**
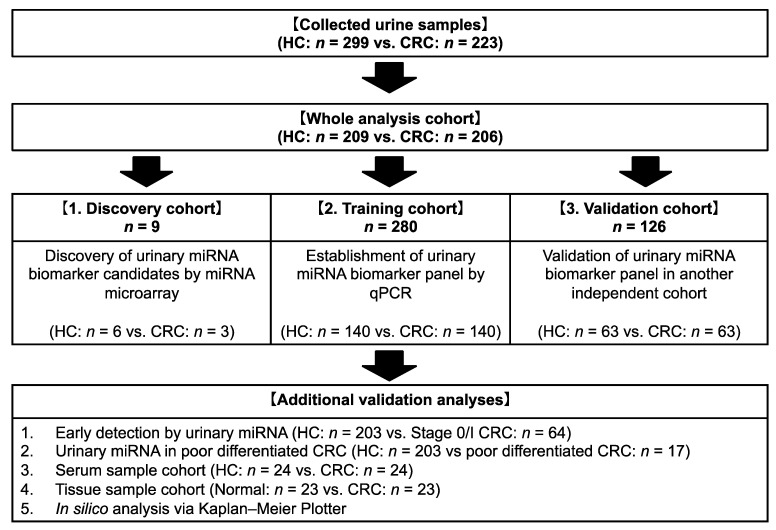
Study profile. HC, healthy control; CRC, colorectal cancer; qPCR, quantitative polymerase chain reaction.

**Figure 2 cancers-14-00461-f002:**
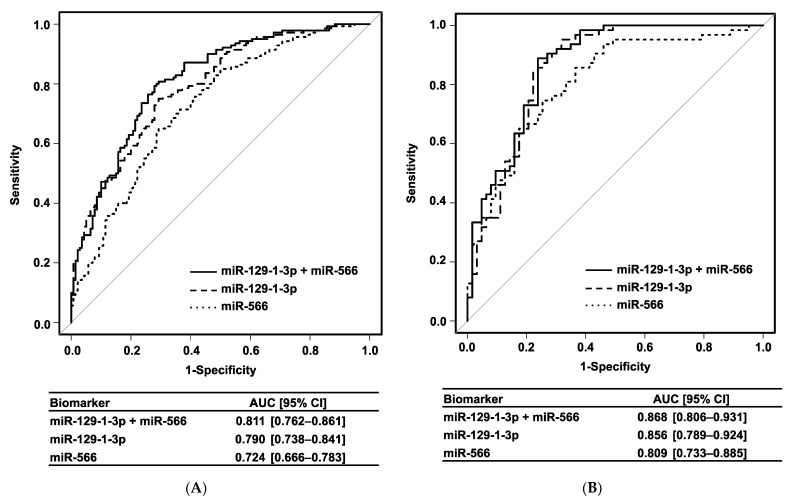
Receiver operating characteristics curves. (**A**) Training cohort. (**B**) Validation cohort. AUC, area under the curve; 95% CI, 95% confidence interval.

**Figure 3 cancers-14-00461-f003:**
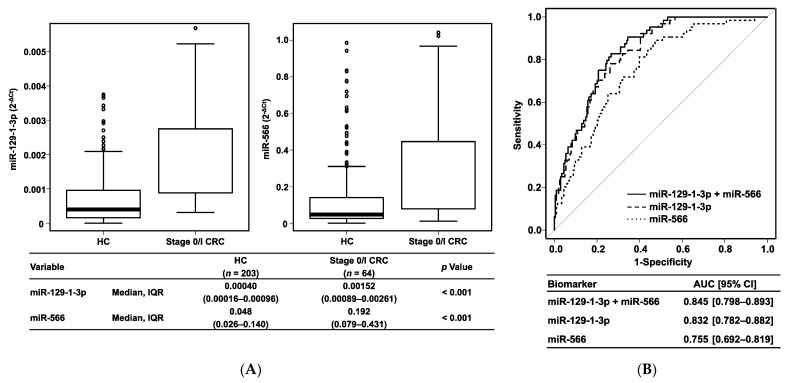
Urinary miRNA biomarker in stage 0/I CRC. (**A**) Boxplots. (**B**) Receiver operating characteristics curves. HC, healthy control; CRC, colorectal cancer; IQR, interquartile range; AUC, area under the curve; 95% CI, 95% confidence interval.

**Table 1 cancers-14-00461-t001:** Patient characteristics.

Item		HC	CRC		*p* Value
		(*n* = 209)	(*n* = 206)		
Age (years)	Median (IQR)	69 (63–74)	69.5 (63–75)		0.275
Gender, *n*	Male	123	117		0.672
	Female	86	89		
Serum Cr (mg/dL)	Mean ± SD	0.78 ± 0.18	0.77 ± 0.26		0.787
Histological grade, *n*, %	well to mod		189	(91.7)	
	por		17	(8.3)	
Location, *n*, %	Cecum		20	(9.7)	
	Ascending		29	(14.1)	
	Transverse		22	(10.7)	
	Descending		8	(3.9)	
	Sigmoid		54	(26.2)	
	Rectum		73	(35.4)	
Stage, *n*, %	0		22	(10.7)	
	I		44	(21.4)	
	II		44	(21.4)	
	III		48	(23.3)	
	IV		48	(23.3)	

HC, healthy control; CRC, colorectal cancer; IQR, interquartile range; Cr, creatinine; SD, standard deviation; well to mod, well to moderately differentiated adenocarcinoma; por, poorly differentiated adenocarcinoma.

**Table 2 cancers-14-00461-t002:** Urinary miRNA expression in the training phase.

Variable	2^−ΔCt^ (Median, IQR)		Univariate	Multivariate		
			Analysis	Analysis		
	HC	CRC	*p* Value	Odds Ratio (95% CI)		*p* Value
	(*n* = 140)	(*n* = 140)				
miR-129-1-3p	0.00054	0.00150	<0.001	5.59	(2.82–11.10)	<0.001
	(0.00019–0.00101)	(0.00082–0.00326)				
miR-566	0.050	0.184	<0.001	1.64	(1.09–2.45)	0.017
	(0.029–0.163)	(0.071–0.438)				
miR-598-5p	0.122	0.273	<0.001			
	(0.070–0.266)	(0.130–0.402)				

IQR, interquartile range; 95% CI, confidence interval.

**Table 3 cancers-14-00461-t003:** Urinary miRNA biomarker in the validation cohort.

Variable	2^−ΔCt^ (Median, IQR)		Univariate	Multivariate		
			Analysis	Analysis		
	HC	CRC	*p* Value	Odds Ratio (95% CI)		*p* Value
	(*n* = 63)	(*n* = 63)				
miR-129-1-3p	0.00025	0.00141	<0.001	5.03	(1.99–12.70)	<0.001
	(0.00015–0.00079)	(0.00095–0.00218)				
miR-566	0.040	0.222	<0.001	2.99	(1.13–7.89)	0.027
	(0.017–0.105)	(0.100–0.526)				

IQR, interquartile range; 95% CI, confidence interval.

## Data Availability

The data presented in this study are available on request from the corresponding author (T.S.).

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
