# Peer review of "A Novel Urinary miRNA Biomarker for Early Detection of Colorectal Cancer"

_cancers, 2022, doi:10.3390/cancers14020461_

Round 1
Reviewer 1 Report
Authors in the paper revealed some miRNAs in the urine samles as biomarkers of colorectal cancer. The approach is quite interesting and they enrolled a consideable group of patients to the study set. I would like to address some commnets to authors:
- The major point that should be considered is the correlation between blood and urinary miRNAs expression. I think that in the urinary system miRNAs are more affected to be disintegrated. It will be very interesting if authors correlate level of the selected miRNAs in blood and urine even in small cohort of patients. They will consider based of the mentioned that urin is a better specimen because is obtained non-invasivelly and reflect the body miRNAs status.
- Please correlate the miRNAs expression with clinical-demographic data of patients. It could be presented in a table.
- It will be interesting to more carefully handle with data for arly and advanced stages of CRC. For instance, please compare urinary miRNAs expression throughout all tumor stages on a one graph.
- Do the miRNAs correlated with either survival or disease free survival?
Reviewer 2 Report
I find the study relevant and could be of easy application to the clinical practice once validated by independent groups and in independent cohorts. My only concern is related with the high deviation in miRNA levels in the HC group. This, although it seems that miRNA 129 and 566 are very sensitive in detecting CRC patients, they will most likely produce a high number of false positives. Because I'm not an expert in statistics I will leave this issue open for discussion.
Reviewer 3 Report
This is an interesting study and the findings are important for applying an non-invasive molecular biomarker for screening of CRC patients, so that they can be identified at earlier stage. However, I think the authors need to clarify the following points to ensure the precision of study design.
Firstly, how was the HC patients recruited in this study? Did they show any symptoms for colorectal disease? Were both CRC and HC candidates followed the same urine collection protocol, and recruited synchronously from all the three institutions, or HC candidates from institution other than that recruited CRC patients? Was there any criteria of time point for CRC and HC candidates to provide urine samples (for example, the first urination in the morning)?
Did the HC and CRC candidates received test like FOBT, FIT, etc? How was the results compared to this urine miRNA biomarker?
Was the same formula derived from log regression model used in both training and validation set?
To further ensure the performance of this urine miRNA biomarker for CRC diagnosis, the authors should perform a prospective study to determine its predictive potential.
Round 2
Reviewer 1 Report
Authors correctly responded to all addressed comments. I have no additional comments.
Author Response
Authors correctly responded to all addressed comments. I have no additional comments.
We thank favorable comment for this Reviewer .
Reviewer 3 Report
Thanks for your reply to my previous comments.
Further to my first concern, I wonder if the CRC patients received instruction of bowel preparation before collection of urine?
I think it is important for a biomarker study to have validated the cut-off value and formula, and evaluate its performance in a prospective study. So to my opinion I expect this part can be included in this manuscript.
Round 3
Reviewer 3 Report
Though this study is interesting, the current design (lack of prospective study, formula used in each stage is different) is not solid enough to support the performance of this urinary miRNA biomarker as diagnostic biomarker.